# Transcriptome Analysis of Seed Weight Plasticity in *Brassica napus*

**DOI:** 10.3390/ijms22094449

**Published:** 2021-04-24

**Authors:** Javier Canales, José Verdejo, Gabriela Carrasco-Puga, Francisca M. Castillo, Anita Arenas-M, Daniel F. Calderini

**Affiliations:** 1Institute of Biochemistry and Microbiology, Faculty of Sciences, Universidad Austral de Chile, 5110566 Valdivia, Chile; castillo.francisca88@gmail.com (F.M.C.); ana.arenas@uach.cl (A.A.-M.); 2ANID–Millennium Science Initiative Program-Millennium Institute for Integrative Biology (iBio), 8331150 Santiago, Chile; 3Graduate School, Faculty of Agricultural Sciences, Universidad Austral de Chile, 5110566 Valdivia, Chile; jose.verdejo.a@outlook.com; 4Plant Production and Plant Protection Institute, Faculty of Agricultural Sciences, Universidad Austral de Chile, 5110566 Valdivia, Chile; gacarcl@gmail.com

**Keywords:** *Brassica napus*, seed weight, seed number, gene co-expression, network analysis, transcriptomics, source–sink

## Abstract

A critical barrier to improving crop yield is the trade-off between seed weight (SW) and seed number (SN), which has been commonly reported in several crops, including *Brassica napus*. Despite the agronomic relevance of this issue, the molecular factors involved in the interaction between SW and SN are largely unknown in crops. In this work, we performed a detailed transcriptomic analysis of 48 seed samples obtained from two rapeseed spring genotypes subjected to different source–sink (S–S) ratios in order to examine the relationship between SW and SN under different field conditions. A multifactorial analysis of the RNA-seq data was used to identify a group of 1014 genes exclusively regulated by the S–S ratio. We found that a reduction in the S–S ratio during seed filling induces the expression of genes involved in sucrose transport, seed weight, and stress responses. Moreover, we identified five co-expression modules that are positively correlated with SW and negatively correlated with SN. Interestingly, one of these modules was significantly enriched in transcription factors (TFs). Furthermore, our network analysis predicted several NAC TFs as major hubs underlying SW and SN compensation. Taken together, our study provides novel insights into the molecular factors associated with the SW–SN relationship in rapeseed and identifies TFs as potential targets when improving crop yield.

## 1. Introduction

Research on the physiological and molecular clues in controlling seed weight (SW) has increased in recent years in model plants and staple food crops, driven by the challenge of developing high-yield varieties in order to attain food security and quality seed traits under sustainable production systems [1]. Therefore, gains in genetic yield potential are urgently needed to meet growing demands [2]. Globally, rapeseed (*Brassica napus* L.) is the third most important oilseed crop, after palm and soybean, used for oil production, animal feed, and biofuel [3]. In oil crops, the rising demand for biofuels has resulted in exponential growth, reaching an estimated global production of 612 million Mg across oil crops in 2018 [4]. This growth has also been evident in rapeseed production since 1980, reaching 75 million Mg in 2019 [5]. However, the stagnation of the yield in rapeseed has been shown in countries such as the United Kingdom, Brazil, Finland, Sweden, and the Czech Republic [6,7,8]. Therefore, new strategies aimed at increasing seed yield are urgently needed. To this end, an integrated approach combining physiology and gene mining knowledge should allow us to uncover key associations that determine complex traits such as yield.

Seed weight is a conservative plant trait across grain crops, showing high heritability [6,9,10]. In addition to these features, the SW of rapeseed has also shown narrower plasticity than other crops, such as cereals, e.g., wheat and barley [11], and oil crops, such as sunflower [12,13]. However, it has been recently demonstrated that the SW of rapeseed can fully compensate for decreases in seed number (SN) under source reduction at flowering [14]. In this regard, other work has also shown the compensation, though partial, for the SN decrease through increased SW around flowering [15,16]. This background offers an outstanding opportunity to study the physiological and molecular bases of SW regulation in rapeseed. Different genes have been reported to be involved in SW and seed size determination, such as key genes involved in the regulation of cell size and cell number of seeds, highlighting the ubiquitin–proteasome pathway [17]; phytohormone biosynthesis/signaling [18,19]; transcriptional regulatory factors, such as APETALA2 and MADS-box [20]; sugar signaling [21]. Recently, a detailed transcriptome analysis of two Brassica rapa genotypes with contrasting seed sizes in seven stages of seed development has been reported, identifying a group of cell cycle-related genes connected to variation in seed size [22]. However, these studies have mostly focused on seed size or SW under controlled conditions and, more importantly, without the evaluation of the key SN–SW compensation shown by this crop.

Brassicas are the most closely related crops to the model plant species Arabidopsis thaliana, one of the most extensively studied species in the world. The sequencing of the genome of several Brassicas has provided a key opportunity to harness the rich knowledge obtained in Arabidopsis and to transfer it to staple food crops [23].

The evidence that a reduction in SN can be fully or partially compensated by SW support the high plasticity of rapeseed, previously reported [14,15,16]. Therefore, in order to understand the SW plasticity in rapeseed, an integrated physiological and molecular approach was carried out in this study using the experiments with spring rapeseed that we previously reported [16]. In that work, we found that the source reduction from the beginning of flowering to 15 days after flowering (DAF) by shading added a new scenario to study the determinants of seed plasticity because the thousand seed weight (TSW) was enhanced, ranging from 15 to 39% in response to the seed number (SN) decrease of between 37 and 49% [16]. Therefore, the reduction in SN was partially compensated for by SW, supporting the high plasticity of SW under source reduction previously reported [14,15,16]. Finally, our study aimed to use bioinformatics analysis to identify the genes associated with SN and SW compensation under field conditions, which may be helpful targets for yield improvement in rapeseed and other crops, breaking the extensively reported trade-off between the two major yield components. The feasibility of breaking this trade-off has been recently demonstrated in wheat [24].

## 2. Results

### 2.1. Multivariate Analysis of RNA-Seq Data Uncovers the Main Factors Affecting Seed Transcriptomes in Rapeseed under Field Conditions

In order to gain new insights into the molecular factors underlying SW plasticity in rapeseed under field conditions, we performed a detailed transcriptomic analysis of seed samples obtained from two spring genotypes (Lumen and Solar) subjected to different S–S ratios [16]. We extracted total RNA from seeds at two different times during seed filling to capture early and late responses to S–S ratio treatments (7 and 14 days after flowering, respectively). We obtained 1240 million reads from 48 samples comprising both genotypes, two S–S ratios from two sowing date treatments with three replicates. On average, each sample had 26 million reads, of which 73% were mapped to the Brassica napus reference transcriptome using kallisto software [25] (Appendix A). To interpret this large transcriptomics dataset, we designed a multivariate linear model testing whether the expression of a given gene could be explained by the seed response to the S–S ratio (SS), genotype (G), developmental time (T), sowing date (SD), and the interaction of these factors. We fit all expressed genes with this full linear model using sleuth [26] (Figure 1A), and we found that the expression of 78.7% of the regulated genes (27,353 of 34,764 genes, Appendix A) could be best explained by a single term. This indicates a low synergistic effect of the factors analyzed (Figure 1A). The only remarkable synergistic effect was found among G, T, and SD (4741 genes, Figure 1A and Appendix A). In contrast, the response to the S–S ratio occurred independently from the sowing date (Figure 1A).

Then, we intersected all the significantly regulated genes uncovered by multivariate analysis to find the genes exclusively regulated by each factor. In addition, we filtered out all genes that were regulated by sowing date to reduce the environmental effect on the main factors. In this manner, we identified 9227, 6698, and 1014 genes exclusively regulated by T, G, and SS, respectively (Figure 1B and Appendix A). The number of genes affected by two or more factors was low except for the cases of SS and T, which were higher than SS alone (Figure 1B).

### 2.2. Genes Regulated by Development Time under Field Conditions Are Highly Enriched in Biological Processes Related to Seed Filling

The experimental factor with a higher impact on the rapeseed transcriptome of seeds was developmental time. A detailed analysis of these genes is relevant because it provides information about the biological processes that are regulated during seed filling under field conditions. To identify genes preferentially expressed at the initial and middle stages of seed development, we performed a hierarchical clustering analysis of the genes exclusively regulated by time using the Pearson’s correlation to measure the similarity among genes. In this manner, two clusters with similar sizes were identified (Figure 2). Cluster 1 is composed by 4173 genes for which the expression was higher at 7 DAF. This temporal expression pattern was consistent between genotypes, S–S ratio treatments, and sowing dates. To provide an overview of the biological pathways related to the genes of cluster 1, we performed a Gene Ontology (GO) overrepresentation analysis using BiNGO software [28]. We found that ribosomal assembly, histone modification, response to cadmium ion, and ovule development were the most enriched biological processes (Figure 2 and Appendix A). Genes encoding for ribosomal proteins of the L10 and L34 families were expressed in higher levels at 7 DAF compared with at 14 DAF (Appendix A). In addition, this set of genes was significantly enriched (q-value < 0.01) in cell division-related GO terms including “DNA unwinding involved in replication”, “regulation of DNA metabolic process”, and “cell division” (Figure 2 and Appendix A). In summary, the functional overview of these genes suggests that protein biosynthesis and cell division are biologically active functions during the early stages of seed development.

On the other hand, the expression levels of genes from cluster 2 were increased at 14 DAF. The most enriched biological processes were related to vacuole organization and Golgi vesicle-mediated transport (Appendix A). In addition, we detected other significantly enriched GO terms related to lipid metabolism, such as the “acetyl-CoA biosynthetic process” or the “positive regulation of fatty acid biosynthetic process”. These results suggest that genes related to lipid biosynthesis and storage protein trafficking are preferentially expressed during the mid-stages of seed development.

### 2.3. Reduction in the Source–Sink Ratio at Flowering Induces the Expression of Genes Involved in Stress Response and Seed Weight

Manipulations of the S–S ratio at the beginning of seed filling is a well-known experimental factor that impacts the seed yield components in rapeseed [14,15,16,30]. We previously showed that the shading treatment during seed filling reduces SN and SW in rapeseed [14,16]. To obtain information about the molecular factors underlying the negative effect of reducing the S–S ratio on SN and the positive impact on SW, we analyzed the expression patterns and functional annotations of 1014 genes exclusively affected by the shading treatment in more detail. Hierarchical clustering analysis showed that 90.4% of these genes (917 out 1014 genes) were induced by shading and that only 97 genes were downregulated by this treatment (Cluster 2 and Figure 3A). A Gene Ontology enrichment analysis reveals that several biological functions related to stress response, including the ABA signaling pathway were induced by the S–S ratio (Figure 3A and Appendix A). Moreover, we found sucrose transport between the most overrepresented biological processes, which includes genes associated with seed weight such as bidirectional sugar transporters such as *SWEET 11* and *SWEET 12*. We selected three genes of the GO term “sucrose transport” to show the regulatory effect of shading treatment on the expression of these genes (Figure 3B). Interestingly, we found that genes regulated by the S–S ratio significantly intersect (*p*-value < 0.05) a list of well-known genes involved in SN and SW determination [31], which include *SWEET* genes, receptor-like protein kinase *FERONIA* (*FER*), or the RING-type protein with E3 ubiquitin ligase activity (*DA2*) [32] (Figure 3C).

### 2.4. Gene Co-Expression Network Analysis Uncovers Novel Gene Modules Related to Seed Yield and Quality under Field Conditions

In order to identify the key molecular drivers underlying important agronomic traits, such as SN, SW, or oil concentration, we performed a weighted gene co-expression network analysis (WGCNA) [33] with all of the expressed genes at two different developmental times for rapeseed seeds and the physiological traits obtained from the same experiment [16] (Appendix A). WGCNA allows for the identification of modules of highly co-expressed genes and relates the modules to external sample traits [33]. We excluded genes for which the mRNA levels were consistently low (less than 5 tpm in more than 90% of the samples) from this gene co-expression network analysis. In addition, we filtered out 25% of the genes with the lowest coefficients of variation at each developmental time since non-varying genes are less likely to have biologically relevant differences. In total, 25,963 and 27,025 genes meet these criteria at 7 and 14 DAF, respectively. After performing the WGCNA analysis with these genes, we obtained a gene co-expression network composed of 27 different modules at 7 DAF (Figure 4 and Appendix A) and a second network with 17 different modules at 14 DAF (Figure 5 and Appendix A). Therefore, the number of modules detected at 14 DAF was remarkably lower than that during the early stage of seed development (7 DAF), suggesting that the transcriptome complexity is higher during initial stages of the seed filling period.

In the 7 DAF network, we found that 11 out of 27 modules had a significant correlation (*p*-value < 0.01) with at least one physiological trait (Figure 4A). Nonsignificant correlated modules with physiological traits were discarded for further analyses. The turquoise, blue, yellow, and red modules were the biggest modules, with more than 500 genes each (Figure 4B). Interestingly, the yellow and red modules were significantly correlated with SN and TSW, whereas the turquoise and blue modules were correlated with seed and protein content.

An inverse relationship between SN and TSW was observed in the first case. The same inverse relationship was obtained for the second case. The GO enrichment analysis revealed that all of the modules correlated with physiological traits contained overrepresented biological processes (q-value < 0.01) except for the case of the salmon and cyan modules (Figure 4B). An intersection analysis showed that more than 80% of genes from the turquoise and blue modules were significantly regulated by genotype according to the multivariate analysis (Figure 4C). The yellow module was the only co-expression module that showed a high percentage of genes regulated by the S–S ratio treatment (>80%), whereas several small modules, such as green-yellow, cyan, or light green, were enriched in genes regulated by developmental time and sowing date (Figure 4C).

In the case of the gene co-expression network constructed from the dataset of 14 DAF, we identified 12 out of 17 modules showing significant correlation (*p* < 0.01) with at least one physiological trait (Figure 5A). The GO enrichment analysis showed that eleven modules were enriched (q-value < 0.01) in GO terms related to specific biological functions. The average size of these modules was 1148 genes, which is 1.6-fold higher than that in the early developmental time network. In fact, the number of modules with more than 500 genes was twice that in the previous time network (8 vs. 4) (Figure 5B). The gene co-expression modules of larger sizes (turquoise and blue) was significantly correlated with seed protein and oil concentration. Similar to the previous gene co-expression network, these modules were also enriched in genes regulated by genotype (Figure 5C).

### 2.5. Identification of Regulatory Factors Associated with the Compensation of SW and SN Decreases by Source–Sink Restriction at Flowering

The 7 DAF co-expression network showed that three modules positively correlated with SW and negatively with SN (yellow, light-green, and cyan; Figure 4). The light-green and cyan modules were discarded after further analysis due to their low enrichment in genes significantly regulated by S–S treatment and their low number of enriched GO terms (Figure 4B). In contrast, the yellow co-expression module showed a high percentage of significantly regulated genes in the S–S treatment (>80%, q-value < 0.01) and 16 significant enriched biological functions (Figure 4, q-value < 0.01). Therefore, we selected this module for further detailed analysis and to identify the candidate genes related to SW compensation. This module was composed of 1023 genes that are associated with several biological processes related to stress and phytohormone responses and signaling (Figure 6). Interestingly, we found that the yellow module is significantly enriched in the molecular function “transcription factor activity” with 96 genes (Figure 6C). The NAC family was the most prevalent family of TFs in this module with 25 genes (Figure 6D). In order to identify candidate TFs underlying SW increase, we used intramodular connectivity since the relationship between connectivity and gene essentiality is well known [34]. Specifically, a module membership (kME) threshold higher than 0.85 was selected as an indicator of high intramodular connectivity to identify hub genes. In addition, candidate TFs were ranked by their average gene significance (GS) with SN and SW traits. In this way, we identified *BnaC09g47170D* (*NAC082*), *BnaA07g38140D* (*CRF6*), and *BnaC01g44850D* (*NGAL2*) at the top TFs (Appendix A). Interestingly, it has been demonstrated that NGAL2 regulates seed size in Arabidopsis [35,36]. Moreover, we found two TFs (NAC041 and IDD1) with high intramodular connectivity, which have been previously associated with cell wall biosynthesis and seed development in Arabidopsis, respectively [37,38]. As shown in Figure 6E, the mRNA levels of *NAC082*, *NGAL2*, *NAC041,* and *IDD1* were positively correlated with SW and negatively correlated with SN (*p*-value < 0.001), indicating that these TFs might be involved in the control of the SW and SN relationship.

In the case of the 14 DAF co-expression network, we found that the red and midnight blue modules were significantly correlated with SW and SN (Figure 5A), although only the red module is highly enriched in genes significantly regulated by the S–S treatment (>80%, Figure 5C). Interestingly, the expression profile of this module is similar to that of the yellow module of the 7 DAF co-expression network (Figure 7A); in fact, 53% of genes from the red module are shared with the yellow one. However, the size of the red module is about 40% lower than that of the yellow module of the 7 DAF co-expression network (638 vs. 1023 genes), suggesting that genes involved in the SW response are preferentially expressed at early stages of seed development. Accordingly, the total number of genes significantly correlated with SW and SN is 1586 genes at 7 DAF, which is two-fold higher than the case of 14 DAF network (739 genes). Overall, these results suggest that early stages of seed development are better predictors of final SW and SN.

In addition, the GO term enrichment analysis reveals that “protein-chromophore linkage”, “response to abscisic acid stimulus”, and “photosynthesis” were the most significantly enriched biological functions associated with the red co-expression module (Figure 7B). To identify TF candidates for this module, we applied the same criteria used for the yellow module of 7 DAF. In this way, *BnaA09g41470D* (*LRL1*), *BnaA05g24050D* (*NAC3*), *BnaA03g05670D* (*bZIP3*), and *BnaC01g02120D* (*HB40*) were identified as top TFs in this module (Appendix A). As shown in Figure 7C, several of these TFs (LRL1, bZIP3, and NAC3) are significantly correlated with SW and SN at 14 DAF, suggesting that these TFs might be SW and SN regulators in *Brassica napus*.

## 3. Discussion

Understanding the molecular mechanisms and genetic factors underlying complex agronomic traits, such as SW and quality, is vital for precise plant breeding [40]. It has been recently shown that transcript levels are useful for predicting complex traits such as plant height, flowering time, and grain yield in maize [41]. In fact, this study shows that transcriptome-based models have better prediction performance than genetic markers in the case of flowering time, suggesting that transcriptome data can provide a link to complex traits that cannot be readily captured at the sequence level [41].

In rapeseed, several transcriptomic studies have been carried out to identify candidate genes associated with seed oil content [42,43,44,45,46]. In contrast, very few studies have focused on the identification of genes associated with SW in *Brassica napus* [31,47]. Moreover, a comprehensive analysis that integrates transcriptome and physiological data under field conditions is still lacking, and even more scarce is information on the compensatory response of SW to a SN decrease. A critical issue for improving crop yield is the compensatory effect between SW and SN that has been observed in *Brassica napus* [14,16]. Despite the agronomic relevance of this issue, the molecular factors involved in the interaction between SW and SN are largely unknown in crops. In this study, we combined transcriptome sequencing with agronomic traits to obtain candidate genes associated with SW and SN under field conditions.

### 3.1. Identification of Genes Related to Seed Weight Plasticity in Rapeseed

A multifactorial analysis of RNA-seq data revealed that most of the significantly regulated genes (q-value < 0.01) were only affected by one experimental factor, with developmental time and genotype being the most influential factors on the *Brassica napus* seed transcriptome under field conditions. Regarding the genes exclusively regulated by time, we found several significantly enriched biological processes related to protein biosynthesis and lipid metabolism at 7 DAF and 14 DAF, respectively. These results are consistent with previous findings, showing that oil biosynthesis is initiated at about 14 DAF in rapeseed [45,48] while protein accumulation begins earlier [49]. These results indicate that our pipeline of multifactorial RNA-seq analysis captures relevant biological information regardless of environmental noise under field conditions.

We have previously shown that shading treatment negatively affects SN and positively affects SW, allowing for key compensation in seed yield [16]. In order to gain further insights into this compensation, we analyzed the expression patterns and functional identity of genes that were consistently regulated by the S–S treatment across genotype, developmental time, and sowing dates. Most of the genes exclusively regulated by the S–S treatment (928 out 1027) were induced by this factor. Interestingly, sucrose transport is one of the most significant enriched GO terms associated with genes regulated by the S–S treatment. Specifically, we identified two members of the SWEET family that were consistently induced by the shading treatment (Figure 3B). Previous functional studies demonstrated that SWEET proteins are key components of sugar translocation to seeds [50,51,52]. Mutation of *AtSWEET11/12/15* in *Arabidopsis thaliana* severely affects seed development, including reduced seed weight, and reduced starch and lipid content [53]. A similar effect has been reported in rice, where the knockout of *OsSWEET11* and 15 genes results in a complete loss of endosperm development [51,54]. Moreover, recent sequencing data from over 800 soybean genotypes revealed that a gene from the SWEET family (*GmSWEET10a*) has been selected and conferred simultaneous increases in soybean seed size and oil content [52].

Another remarkable example of a gene associated with seed weight and induced by S–S treatment is *BnaA01g19380D*, an ortholog of Arabidopsis *DA2*. This gene encodes a ubiquitin ligase that interacts with the ubiquitin receptor DA1 to synergistically regulate seed size in Arabidopsis [32]. Moreover, it has been demonstrated that the downregulation of an ortholog of *DA1* in *Brassica napus*, *BnDA1*, resulted in a 21% increase in seed weight and 13% increase in seed yield per plant under field conditions [55]. A similar phenotype has been described in maize and wheat [56,57], indicating that the biological function of this gene is conserved in angiosperm plants. Moreover, it has been proposed that *ZmDA1* improves the sugar imports into the sink organ and starch synthesis in maize kernels [56]. Therefore, it is possible that *DA2* could also be involved in the regulation of sugar translocation in response to changes in the S–S ratio.

### 3.2. Transcription Factors Associated with SW and SN Response under Field Conditions

Our study aimed to use bioinformatics analysis to identify genes related to agronomic traits with a special focus on SW–SN regulation, which might be a helpful target in yield improvement and yield stability in rapeseed and other crops. Specifically, we performed a WGCNA analysis from 48 RNA-seq samples obtained from seeds at 7 DAF and 14 DAF to identify gene co-expression modules significantly correlated with SW and SN (Figure 4 and Figure 5, respectively). Interestingly, the yellow module of the 7 DAF network contains most of the genes significantly correlated with SW and SN. The most enriched molecular function was “transcription factor activity”, indicating that this co-expression module is associated with the regulation of gene expression. NAC was the most abundant TF family of this module, with 25 members. NAC TFs are involved in diverse signaling and developmental events, including stress responses, senescence, and seed development [58,59]. Interestingly, NAC TFs have been described as controlling for several clues in seed development [59]. For instance, it has been recently shown that *OsNAC25* and *OsNAC26* bind to the promoters of important genes involved in the control of seed size in rice such as GW2, GW5, and DR11 [60]. Moreover, three NAC TFs (*OsNAC020*, *OsNAC023,* and *OsNAC026*) have been associated with SW in rice since their expression profiles during seed development vary among different accessions with contrasting seed size [61]. Taken together, these results suggest that NAC TFs might be relevant regulatory factors of the SW–SN relationship.

Next, we used the intramodular connectivity and trait correlation to identify candidate TFs from the yellow module of the 7 DAF network. An orthologue of Arabidopsis *NAC082*, *BnaC09g47170D*, was the TF with the highest average correlation to SW–SN. *NAC082* belongs to the NAC domain family and was characterized for the first time in the context of xylem vessel differentiation after discovering that a master regulator of this developmental process, *VND7*, interacts with *NAC082* in Arabidopsis [62]. More recently, it has been reported that *NAC082* is a key regulator that connect the ribosomal defects induced by stress and cell proliferation [63,64]. Specifically, these studies showed that several stresses lead to an increase in *NAC082* expression, which blocks tissue regeneration and delays seed germination. Taking into account this previous evidence, the reduction in source–sink ratio by shading treatments during seed filling triggers the expression of *NAC082,* which may block cell proliferation at early stages of seed development and may lead to a reduction in SN.

It is important to note that several key regulators of seed size are also present at the top of our ranking of highly correlated TFs with SW–SN in the yellow module of the 7 DAF network. For instance, the orthologous gene of *BnaC01g44850D* in Arabidopsis, *NGAL2* (third gene in Appendix A), has been shown to regulate seed size by restricting cell proliferation in the integuments of ovules and by developing seeds [35]. The overexpression of *NGAL2* dramatically decreases the seed size of Arabidopsis wild-type plants, whereas the disruption of this gene causes large seeds [35]. In fact, a gain-of-function mutant of the *NGAL2* gene was identified as a suppressor of the large seed phenotype of *DA1* [35], which is a key ubiquitin receptor for seed size determination [65]. Another TF involved in seed development identified in this co-expression module is *IDD1* (orthologous to the rapeseed *BnaA09g52630D* gene). The seeds of the Arabidopsis *IDD1* over-expressor lines were larger than those of wild-type plants due to an enlarged endosperm, higher seed weight in all the stages before seed maturity, and a significant delay in seed development [37].

Taken together, our study provides new insights into the molecular factors that regulate the SW and SN interaction in rapeseed. We identified several new TFs that have not been previously associated with SN and SW and were co-expressed with key regulators of seed size, such as *NGAL2*. The characterization of these TFs may improve the understanding of the regulatory mechanisms underlying the interaction between SW and SN in rapeseed and other crops, creating new pathways for crop yield improvements and yield stability via SW compensation to environmental conditions affecting SN.

## 4. Materials and Methods

### 4.1. Field Experiment, Treatments, and Crop Management

The field experiment and conditions were described in detail by Verdejo and Calderini [16]. Briefly, the field experiments on two sowing dates were carried out at the Austral Farming Experimental Station in Valdivia, Chile (39°47′ S, 73°14′ W). In this experiment, three sources of variation were evaluated in rapeseed: (i) genotype (two adapted spring rapeseed hybrids: Lumen and Solar (chosen for their similar phenology and adapted to southern Chile)), (ii) sowing date (optimal and late sowing dates), and (iii) source–sink (S–S) ratios (control without manipulation and a reduced S–S_ratio_ with black nets intercepting 75% of solar radiation from the beginning of flowering (BBCH 61) to 15 days after flowering (DAF)). The treatments were arranged in a split–split plot design with three replicates, where the sowing date was assigned to main plots, the S–S_ratio_ was assigned to subplots, and the genotypes were assigned to sub-sub plots [16]. The rapeseed plants were sown at a plant density of 55 plants m^−2^.

### 4.2. Phenology and Physiological Plant Sampling

Crop development was followed twice a week in both experiments according to the BBCH phenological scale for rapeseed [66]. In order to determine the SN, TSW, and quality traits as seed oil and protein concentration, the seed samples were harvested in one lineal meter from the central rows of each plot at maturity (BBCH 89), when the seeds inside the siliques were dark and hard. The seed number was measured after oven drying the samples at 65 °C for 48 h using a seed counter (Pfeuffer GmbH, Kitzengen, Germany). Then, seed yield was measured and TSW was estimated as the ratio between seed yield and SN.

The oil concentration of seeds was determined by near infrared reflectometry (NIR) (Foss Infratec 1241, Hilleroed, Denmark) and the nitrogen concentration of seeds was measured using the Kjeldahl procedure [67]. The protein concentration of the seeds was calculated with a conversion factor of 5.8 [68]. The concentrations of both oil and protein are expressed on a dry matter basis.

### 4.3. Seed Sampling and RNA Isolation

In spring rapeseed, SW plasticity is maximized by shading within a time window from 0 to 15 days after the start of flowering [16]. Therefore, in order to elucidate the relevant modules or genes involved in SW determination, we chose two development stages to perform RNA-seq from seeds: in the middle and at the end of the source–sink treatments. The seed samples for the RNA-seq were collected from 5 plants per plot (25 siliques per plant from the bottom of the main raceme) in two development stages: 7 and 14 after the beginning of the shading treatment in three replicates across the different factors analyzed (genotypes, S–S ratios, and sowing dates). The seed samples were fast frozen in liquid nitrogen and then kept at −80 °C until they were processed. The seeds were gridded using liquid nitrogen and a cold mortar and pestle. Total RNA extraction was performed using a method adapted for *Brassica* seeds [69] with 100 mg of each sample and using PCR mini columns (NucleoSpin^®^ Gel and PCR columns; Macherey-Nagel, Düren, Germany) in accordance with previously described protocols [70].

### 4.4. RNA-Seq Analysis

The RNA samples were shipped in dry ice to Novogene facilities in Sacramento, CA USA. RNA quality analysis, library construction, and sequencing were performed by Novogene (Beijing, China). The samples were sequenced using a 2  ×  150 bp kit on an Illumina Novaseq 6000 aiming for 6 Gb per sample. The average Q20 per sample was 97.7%, and that for Q30 was 93.8%.

RNA-seq data analysis was performed as described by [71]. Briefly, sequenced reads were pseudo-aligned to the publicly available *Brassica napus* transcriptome obtained from *Ensembl Plants* using kallisto (v0.46) [25]. The transcript indices for kallisto were generated from rapeseed annotation version AST_PRJEB5043_v1, which includes 101,040 cDNAs (https://plants.ensembl.org/Brassica_napus/, accessed on 23 March 2021). A multivariate linear model to test whether the expression of a given gene could be explained by the S–S ratio, genotype, sowing date, or the interaction of these factors using the R package sleuth (v.0.30.0) [26]. This R package was also used to obtain the normalized expression data in transcripts per million (Appendix A). In order to identify significantly regulated genes for each factor, we applied a q-value threshold of 0.01. The intersection of the gene lists and their statistical significance were assessed using the SuperExactTest R package [27].

### 4.5. Gene Co-Expression Network Construction

Co-expression networks were built for two separate sample sets (7 and 14 DAF) using the method of weighted gene co-expression network analysis (WGCNA) [33]. For each network, we filtered out genes for which the counts were consistently low, with less than 5 tpm in more than 90% of the samples. The count expression level of each gene was normalized using the method implemented in sleuth [26]. The soft-power threshold was chosen as the first power to exceed a scale-free topology fit index of 0.8 [72] for each network. The soft powers that fulfil this criteria were 9 for the 7 DAF network and 8 for the 14 DAF network. Then, the co-expression matrix was calculated using these power values, with a minimal module size of 30 and a merge cut height of 0.2 without PAM stage. Module–trait relationships were estimated using the association between the module eigengenes and the agronomic traits. For each expression profile, the gene significance (GS) was calculated as the absolute value of the association between the expression profile and each agronomic trait.

### 4.6. Gene Ontology Enrichment Analysis

Gene ontology (GO) terms for all of the *B. napus* genes were assigned based on *A. thaliana* orthologous genes according to *Ensembl Plants* in order to gain more informative enrichment results. BiNGO was then used to identify the significantly enriched GO terms (adjusted *p*-value < 0.05) using a hypergeometric test [28], and redundancy between GO terms was reduced using REVIGO [73].

## Figures and Tables

**Figure 1 ijms-22-04449-f001:**
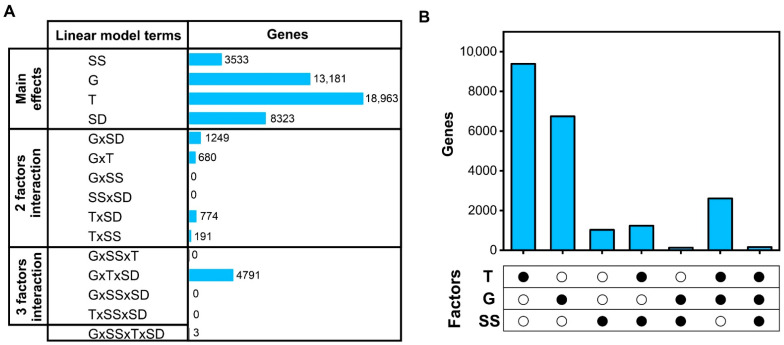
Multivariate analysis of the RNA-seq data obtained from seeds of two rapeseed hybrids grown under different source–sink ratios. (**A**) Multifactorial analysis showing how many genes were regulated by the S–S ratio (SS), genotype (G), developmental time (T), sowing date (SD), and the interaction of these factors (q-value < 0.01). An average of 26 million reads per sample were pseudo-aligned to the *Brassica napus* reference transcriptome using kallisto [25] and a fully mapped dataset was multivariate analyzed with the sleuth R package [26]. (**B**) An intersection analysis between genes regulated by the S–S ratio (SS), genotype (G), and developmental time (T). Genes significantly affected by SD or factors interactions were discarded from this analysis, which was performed using the SuperExactTest R package [27]. The black points indicate which factors affect the expression levels of the genes shown in the bar diagram.

**Figure 2 ijms-22-04449-f002:**
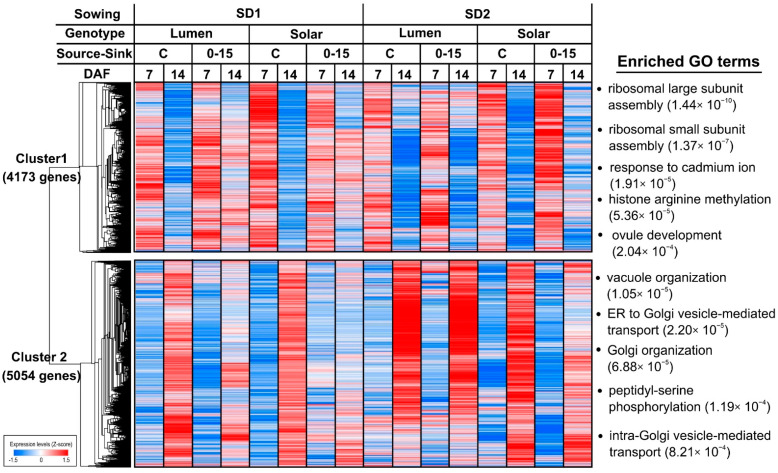
Genes exclusively regulated by developmental time are associated with seed filling. A heatmap showing the two major expression patterns of genes exclusively regulated by developmental time. Hierarchical clustering was performed based on Pearson correlation distances and average linkage using Morpheus software [29]. Each column of the heatmap represents the average expression of three biological replicates. The gene expression values for each gene were normalized by Z-score transformation. The top 5 enriched GO terms of the biological process domain are represented on the right side of each cluster. GO term enrichment analysis was performed by a hypergeometric test using BiNGO software [28]. FDR-corrected *p*-values are indicated for each GO term in brackets. SD = sowing date, DAF = days after flowering, C = control, 0–15 = source to sink treatments performed from the beginning of flowering to 15 DAF.

**Figure 3 ijms-22-04449-f003:**
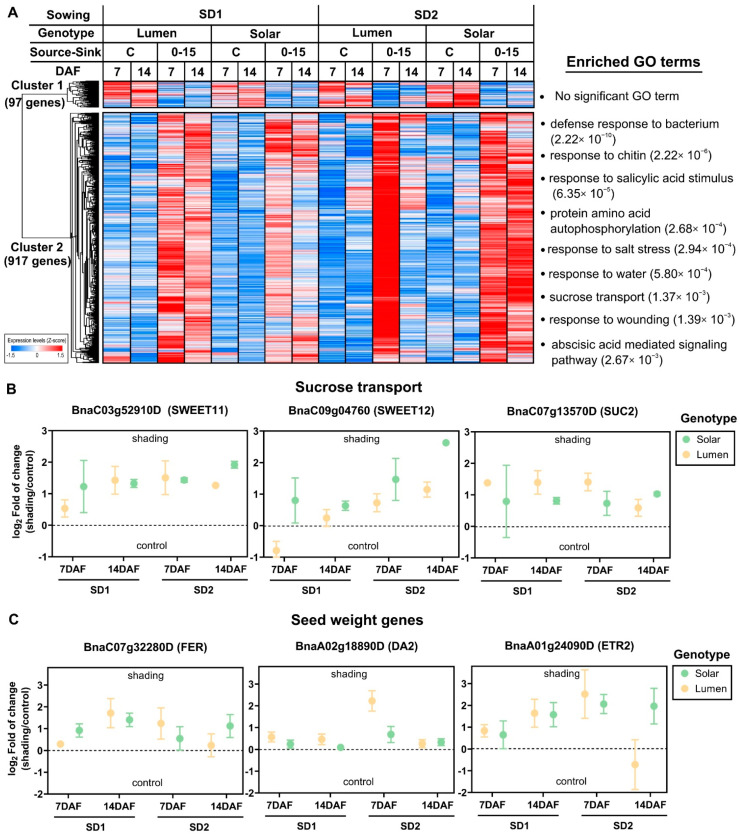
Reduction in the source–sink ratio during seed filling induces the expression of genes involved in sucrose transport, seed weight, and stress response. (**A**) A heatmap showing the two major expression patterns of genes exclusively regulated by shading treatment. Hierarchical clustering was performed based on Pearson correlation distances and average linkage using Morpheus software [29]. Each column of the heatmap represents the average expression of three biological replicates. The gene expression values for each gene were normalized by Z-score transformation. The significant and nonredundant GO terms (adjusted *p*-value < 0.05) of the biological process domain are represented on the right side of each cluster. GO term enrichment analysis was performed by a hypergeometric test using BiNGO software [28]. FDR-adjusted *p*-values are indicated for each GO term in brackets. (**B**) Expression profiles of three representative genes belongs to the GO term “sucrose transport”. The dots represent the average log_2_ fold change between the shading and control samples obtained from normalized RNA-seq data, whereas bars indicate the standard error of the mean (SEM) of three replicates. The Arabidopsis ortholog of each rapeseed gene is indicated in brackets. (**C**) Expression profiles of three representative genes for which the orthologs in Arabidopsis are involved in seed weight regulation. The dots represent the average log_2_ fold of change between shading and control samples obtained from normalized RNA-seq data, whereas bars indicate the standard error of the mean (SEM) of three replicates. The Arabidopsis ortholog of each rapeseed gene is indicated in brackets.

**Figure 4 ijms-22-04449-f004:**
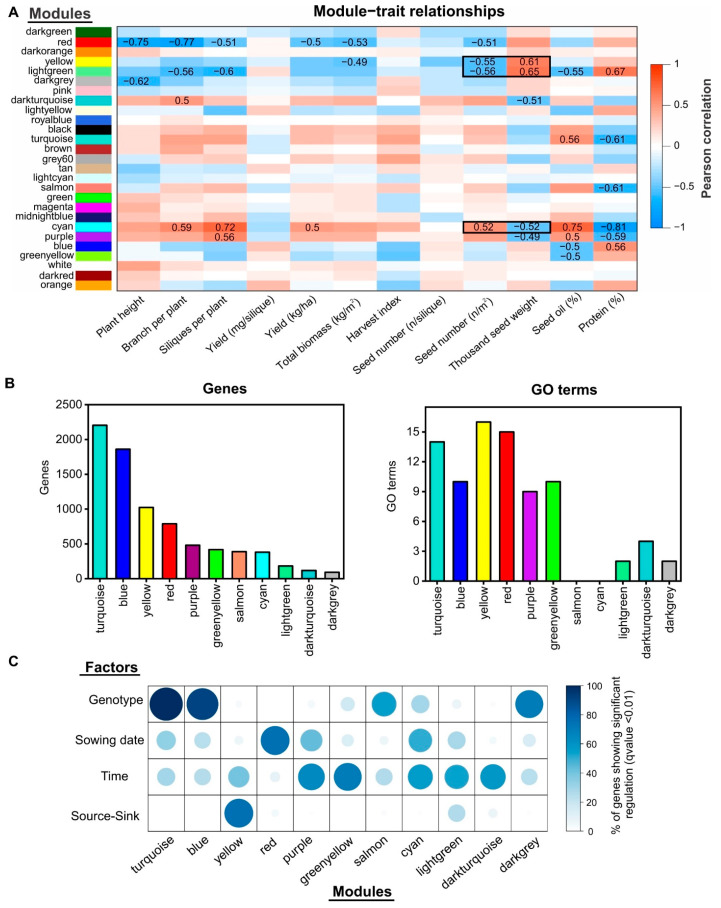
Weighted gene co-expression network analysis identified the modules associated with the SW and SN relationship at 7 DAF. (**A**) A heatmap showing the module–trait associations. Each row corresponds to a module eigengene, and each column corresponds to a trait. Only significant correlations (*p*-value < 0.01) are shown with numbers. Red and blue denote positive and negative correlations with gene expression, respectively. The agronomic traits were obtained from our previous work [16]. Modules significantly correlated with SW and SN are indicated with a black rectangle. (**B**) Histograms showing the number of genes (left) and GO terms (right) per co-expression module. Only modules significantly correlated with at least one agronomic trait are represented. (**C**) Balloon plot showing the percentage of genes in each module that are significantly regulated by each factor.

**Figure 5 ijms-22-04449-f005:**
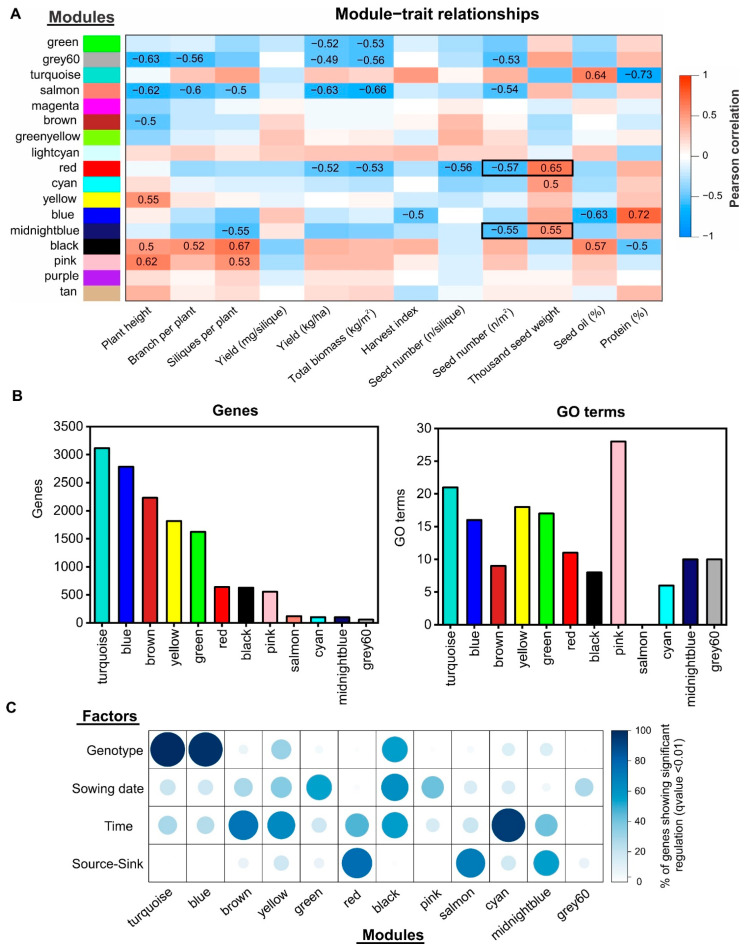
Weighted gene co-expression network analysis identified the modules associated with the SW and SN relationship at 14 DAF. (**A**) A heatmap showing the module–trait associations. Each row corresponds to a module eigengene, and each column corresponds to a trait. Only significant correlations (*p*-value < 0.01) are shown with numbers. Red and blue denote positive and negative correlations with gene expression, respectively. The agronomics traits were obtained from our previous work [16]. The modules significantly correlated with SW and SN are indicated with a black rectangle. (**B**) Histograms showing the number of genes (left) and GO terms (right) per co-expression module. Only modules significantly correlated with at least one agronomic trait are represented. (**C**) Balloon plot showing the percentage of genes in each module that are significantly regulated by each factor.

**Figure 6 ijms-22-04449-f006:**
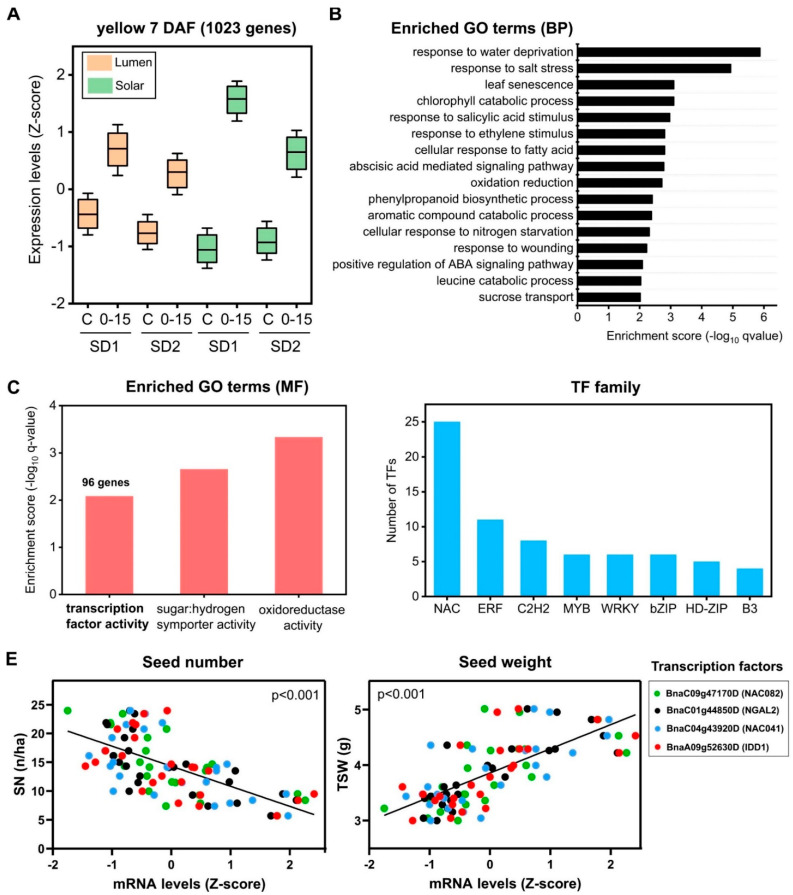
The yellow co-expression module of 7 DAF network is associated with the relationship between SW and SN. (**A**) Expression profiles of genes belonging to the yellow module. On each box, the central mark indicates the median, and the bottom and top edges of the box indicate the 25th and 75th percentiles, respectively. The whisker indicates the standard deviation of the expression data of all genes belonging to the yellow co-expression module. (**B**) GO term enrichment analysis of genes belonging to the yellow co-expression module performed by a hypergeometric test using the BiNGO software and the biological process domain [28]. (**C**) Enriched GO terms of the Molecular Function (MF) domain. The enrichment analysis was performed using the BiNGO software, as indicated above. (**D**) Distribution of the 96 transcription factors of the yellow module according to family. The transcription factors were classified following PlantTFDB4.0 database annotation [39]. (**E**) Relationships between the expression levels of the selected TFs and seed number (right) or seed weight (left). TFs were selected by taking into account their high intramodular connectivity and gene significance with SN and SW traits.

**Figure 7 ijms-22-04449-f007:**
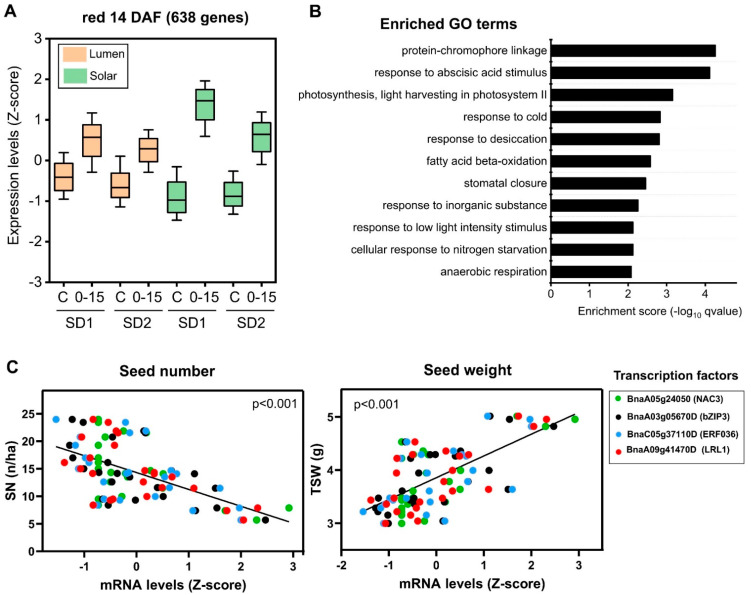
The red co-expression module of the 14 DAF network is associated with the relationship between SW and SN. (**A**) Expression profiles of genes belonging to the yellow module. On each box, the central mark indicates the median, and the bottom and top edges of the box indicate the 25th and 75th percentiles, respectively. The whisker indicates the standard deviation of the expression data of all genes belonging to the yellow co-expression module. (**B**) GO term enrichment analysis of genes belonging to the yellow co-expression module performed by a hypergeometric test using the BiNGO software and the biological process domain [28]. (**C**) Relationships between the expression levels of the selected TFs and seed number (right) or seed weight (left). TFs were selected by taking into account their high intramodular connectivity and gene significance with SN and SW traits.

## Data Availability

The RNA-Seq datasets generated and analyzed during this study are available in the NCBI Gene Expression Omnibus (GEO) repository, accession GSE169511. All other data generated during this study are included in this published article and its Appendix A.

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
