# Peer review of "Transcriptome Analysis of Seed Weight Plasticity in Brassica napus"

_ijms, 2021, doi:10.3390/ijms22094449_

Round 1
Reviewer 1 Report
The manuscript entitled “Transcriptome analysis of the seed weight plasticity in Brassica napus” by Canales et al submitted to IJMS is focused on transcriptomic and bioinformatic analysis of B. napus seed weight plasticity. The present work addresses important agronomical and biological questions regarding crop improvement and the significant role of S-S ratio. The manuscript will be very importnat to the scientific community working on that topic. The manuscript is well written and merits to be published after minor corrections.
I have minor remarks regarding RNA extraction. In line 470-472 the authors wrote : “Total RNA extraction was performed using a method adapted for Brassica seeds [69] with 100 mg of each sample and using NucleoSpin® Gel and PCR columns (Macherey-Nagel, Germany).”
I am curious to know why did the authors used NuceloSpin and PCR columns? I think some extended explanation should be present.
Also in Fig 3 B, C they presented expression profile of 6 selected genes. Did they presented RNA-seq ? I think a validation by qRT-PCR will be worthy.
Overall, I would like to congratulate the authors for their work.
Author Response
Thank you very much for review our manuscript and the overall positive comments.
- Regarding to the first question: “I am curious to know why did the authors used NuceloSpin and PCR columns? I think some extended explanation should be present.”
The main reason to use NucleoSpin Gel and PCR column is that we previously used it for RNA isolation in raps and Arabidopsis, but any silica-based columns may work fine for RNA isolation. For instance, we have also used “PureLink PCR Purification Kit” columns (Invitrogen) in a previous work with good results (Henríquez-Valencia et al., 2018). PCR and plasmid silica-based columns have been demonstrated to yield similar results to columns included in RNA specific kits (https://bmcresnotes.biomedcentral.com/articles/10.1186/1756-0500-5-45), with the advantage to significantly reduce the cost per sample. We have modified the corresponding section to clarify this point (see lines 479-481:” and using PCR mini columns (NucleoSpin® Gel and PCR columns; Macherey-Nagel, Germany) in accordance with previously described protocols [70]”).
2.Regarding to the second question: “Also in Fig 3 B, C they presented expression profile of 6 selected genes. Did they presented RNA-seq ? I think a validation by qRT-PCR will be worthy.”
Yes, we did it. The expression profiles of the selected genes are the log2 FC ratios obtained from normalized RNA-seq data. We have added this information to the caption of Figure 3 (please, see line 219 of the new manuscript version).
We had considered performing qPCR studies to verify some of our gene-expression findings but there is little evidence that qPCR analyses from the same samples will improve our data. Previous studies have shown extremely close correlations between qPCR and RNA-seq data (https://doi.org/10.1038/nmeth.1503;https://doi.org/10.1016/j.gene.2014.01.031; https://doi.org/10.1186/s12870-020-02590-2). To show that an assay is consistent with previous results requires testing a sufficiently large collection of gold-standard examples to be able to assess standard measurements such as sensitivity, false-positive rate and false-discovery rate as was discussed in detail by Hughes (2009) (https://jbiol.biomedcentral.com/articles/10.1186/jbiol104).
Reviewer 2 Report
The paper written by Canales et al. reports on transcriptome analysis of 48 seed samples of rapeseed to identify the factors involved in the interaction between seed weight and seed number. The results show that the reduced source-sink ratio induces the genes for sucrose transport. The authors further identified co-expression modules correlated positively with seed weight and negatively with seed number. Overall, the manuscript and the data are clear and convincing. The only concern I have is that the manuscript contains a lot of grammatical or spelling errors. These should be carefully checked and revised.
For instance,
Lines 187-188, excluded of
Line 196, lower than during
Lines 198, 236 out
Line 226, between for
Line 227, modules contains
Line 241, twice than
Lines 260-261, it is well known the relationship
Line 314, reveal
Line 321, significant
Line 371, regulates
Line 404, after discovered
Author Response
Thank you very much for review our manuscript and the overall positive comments. The new version of the manuscript has been checked by a native English-speaking editor ("Please see the attachment”.)
1.“Lines 187-188, excluded of”. In the new version of the manuscript this mistake has been corrected (see line 191:“excluded from”)
- Line 196, lower than during. In the new version of the manuscript this mistake has been corrected (see line 200:“ lower than that during”)
- Lines 198, 236 out. In the new version of the manuscript we correct this mistake (see line 203: “out of 27 modules” and line 238: “we identified 12 out of 17 modules”)
- Line 226, between for. In the new version of the manuscript we correct this mistake (see line 233: “was obtained for the second case”)
Line 227, modules contains. In the new version of the manuscript this mistake was corrected (see line 234: “contained”)
Line 241, twice than. In the new version of the manuscript this mistake has been corrected (see line 248: “was twice that in the previous time network”)
Lines 260-261, it is well known the relationship. We correct this mistake in this new version of the manuscript (see lines 267-268:” since the relationship between connectivity and gene essentiality is well known”)
Line 314, reveal. In the new version of the manuscript this error has been corrected (see line 319: “GO term enrichment analysis reveals”)
Line 321, significant. In the new version of the manuscript this error has been corrected (see line 285:“Only modules significantly correlated”)
Line 371, regulates. In the new version of the manuscript this error has been corrected (see line 376: “DA1 to synergistically regulate”)
Line 404, after discovered. Done, please see line 409: “vessel differentiation after discovering that a master regulator”.